# Involvement of Estrogen and Its Receptors in Morphological Changes in the Eyes of the Japanese Eel, *Anguilla japonica*, in the Process of Artificially-Induced Maturation

**DOI:** 10.3390/cells8040310

**Published:** 2019-04-03

**Authors:** Ji-Yeon Hyeon, Sung-Pyo Hur, Byeong-Hoon Kim, Jun-Hwan Byun, Eun-Su Kim, Bong-Soo Lim, Bae-Ik Lee, Shin-Kwon Kim, Akihiro Takemura, Se-Jae Kim

**Affiliations:** 1Jeju International Marine Science Research & Logistics Center, Korea Institute of Ocean Science & Technology, 2670 Iljudong-ro, Gujwa, Jeju 63349, Korea; hyeonjy@kiost.ac.kr; 2Department of Biology, Jeju National University, 102 Jejudaehak-ro, Jeju 63243, Korea; sjkim@jejunu.ac.kr; 3Marine Science Institute, Jeju National University, 19-5 Hamdeok 5-gil, Jocheon, Jeju 63333, Korea; endand1011@naver.com; 4Department of Chemistry, Biology and Marine Science, Faculty of Science, University of the Ryukyus, 1 Senbaru, Nishihara, Okinawa 903-0213, Japan; quswns6369@gmail.com (J.-H.B.); takemura@sci.u-ryukyu.ac.jp (A.T.); 5Solforto Co. Ltd., 19 Yeondong 8-gil, Jeju 63133, Korea; skdo1012@daum.net (E.-S.K.); bslim@solforto.com (B.-S.L.); 6Aquaculture Research Division, National Institute of Fisheries Science, 216 Gijanghaean-ro, Gijang, Busan 46083, Korea; bilee5@korea.kr (B.-I.L.); ksk4116@korea.kr (S.-K.K.)

**Keywords:** Japanese eel, *Anguilla japonica*, estrogen receptor, estradiol-17β, ovarian development, eye size, choriocapillary layer, eel spawning

## Abstract

During the long migration from river habitats to the spawning ground, the Japanese eel undergoes sexual maturation. This spawning migration occurs concurrently with morphological changes, such as increases in eye size; however, the mechanisms by which sex steroids and their receptors influence these changes in peripheral tissues remain unclear. The aim of this study was to investigate changes in the eyes of female Japanese eels during sexual maturation, and our research focused on estrogen receptor (ER)α and ERβ transcripts. During ovarian development, the gonadosomatic index increased and yolk-laden oocytes developed rapidly. These changes occurred in conjunction with a steady increase in plasma levels of estradiol-17β (E_2_). Concomitant increases in transcript levels of ERα and ERβ in eye, brain, pituitary, and ovary were also observed. Fluorescence in-situ hybridization analyses revealed that ERα and ERβ transcripts were present in the choriocapillary layer and photoreceptor layer of the eyes, and the analysis also revealed that their signals in these layers became stronger in mature females compared to those observed in immature females, suggesting that under the influence of gonadotropins, morphological changes in the eyes are regulated by E_2_ through the activation of its receptors. In conclusion, E_2_ plays a crucial role in physiological adaptations that occur in peripheral tissues during the spawning migration.

## 1. Introduction

The reproductive activities of female fish are directly and indirectly regulated by sex steroids that are synthesized in the ovarian follicles and secreted into the blood circulation under influence of gonadotropins such as follicle stimulating hormone (FSH) and luteinizing hormone (LH). Among the sex steroids, estradiol-17β (E_2_) is converted from testosterone in the ovarian granulosa cells through the catalytic activity of aromatase [1,2]. Tissue specific actions by E_2_ are mediated by direct binding to intercellular estrogen receptors (ER), which are members of the nuclear receptor superfamily [3,4]. Two ER subtypes (ERα and ERβ) have been reported in mammals [5,6], while three (or more) subtypes, including ERα (alias, ER1, or ESR1), ERβ1 (ER2a or ESR2a), and ERβ2 (ER2b or ESR2b), have been identified in teleosts [7,8,9,10,11]. Elevated transcription of ERs in female fish has been reported in the brain (including the pituitary), liver, and ovary, and changes in the transcript levels of ERs in the liver are closely related to regulation of vitellogenin synthesis in most teleosts [10,12]. ERs within the ovary appear to participate in changes in follicular size in the rainbow trout *Oncorhynchus mykiss* [13]. It has been suggested that the ER-mediated signaling pathway exists in these tissues to mediate E_2_-related reproductive processes [13]. In regard to sexual maturation, however, little attention was paid to the physiological importance of the ER-mediated signaling pathway in the context of other tissues [10]. 

The Japanese eel (*Anguilla japonica*) is a fish species that exhibits a catadromous life cycle and possesses a high commercial value in East Asian countries, where efficient aquaculture is currently undergoing research and development. This species exhibits an extended growing period in freshwater and then undergoes spawning migration to the spawning ground near the Suruga Seamount to the west of the Mariana Ridge [14,15,16]. The sexual maturation of eels occurs during spawning, and it is therefore difficult to examine reproductive processes using individuals in the wild [17]. Difficulty also arises due to the drastic decrease in the natural population of eels, as this animal has recently been listed on the International Union for Conservation of Nature Red List (2014). In this regard, the entire process of reproduction can be replicated using immature eels, as their sexual maturation can be induced by prolonged treatments with salmon pituitary extract (SPE) for females and with human chorionic gonadotropin (HCG) for males [17]. Recombinant gonadotropins specific for Japanese eels are also available for induction of gonadal maturation [18].

In addition to changes in body color from yellow (sexually immature) to silver (sexually mature), sexual maturation of eels is accompanied by an increase in eye size [19,20,21,22,23], suggesting photic perception and adaptation during the spawning migration. It is likely that this morphological change is influenced by reproduction-related hormones, as it has been reported that there is a relationship between 11-ketotestosterone (11-KT) and the eye size of the European eel *A. anguilla* [21]. Despite this, the mechanisms by which E_2_ participates in morphological changes in eyes remain unknown, although ER is transcribed in the eyes of the European eel [11]. The objective of this study was to clarify the involvement of E_2_ in morphological changes that occur in the eyes of the female Japanese eel after intraperitoneal injections of SPE. Our study focused on mRNA expression levels of ER paralogs (ERα and ERβ) in female Japanese eel eyes during the process of sexual maturation using quantitative polymerase-chain reaction (qPCR) and the localization of these paralogs within the eyes using fluorescence in-situ hybridization (FISH).

## 2. Material and Methods

### 2.1. Animals and Hormone Treatment

Wild female Japanese eels were collected using eel traps in the blackish area in Hado, Jeju, Korea (33°N, 126°E) in September 2016. After collection, they were reared without feeding in indoor plastic tanks (1 metric ton capacity) with recirculating freshwater (20 ± 1 °C) under artificial photoperiodic conditions of 12-hours light and 12-hours darkness using fluorescent bulbs (10W, 600 lx, PPFD = 10.0 μmolm^−2^s^−1^, λp = 545 nm) (LD = 12:12, light-on at 06:00 h and light-off at 18:00 h) at the Lava Water Aquatic Animals Care Center in Jeju Techno-Park, Jeju, South Korea. 

Forty-one females (BW: 217–829 g, TL: 54.5–78.5 cm) were transferred to indoor freshwater tanks (5 metric ton capacity) with the filter system and ambient aeration under conditions of photoperiod (approximately LD = 12:12), and with a water temperature at 20 ± 1 °C. Salinity of the tanks was gradually increased to 34.0‰ by addition of seawater for one week. After acclimation to seawater, fish were removed from the tanks and anesthetized with MS-222 (Sigma-Aldrich, St. Louis, MO, USA). Each individual was weighed and tagged with an ID chip. SPE was suspended in saline and intraperitoneally injected (at 20 mg/BW·kg^−1^) once a week (up to 8 weeks). 

Fish (n = 6 to 13 per sampling time) were sampled at 2, 4, 6, and 8 weeks after injections. After anesthetization, fish were sacrificed by decapitation in accordance with the guidelines of the Institutional Animal Care and Experimental Committee of Jeju National University (No. 2016-0039). After weighing, the left side eye, whole brain, pituitary, and ovary were immediately collected, frozen in liquid nitrogen, and stored at −80 °C until analysis. Blood was collected from the caudal vein using a heparinized syringe, transferred into a tube on ice, and then centrifuged at 8000 × *g* for 10 min at 4 °C. Plasma was separated and stored at −80 °C until analysis. Gonadosomatic index (GSI) and eye index (EI) were calculated as follows: GSI = (gonadal mass/body mass) × 100
EI = {[(A + B)/4]^2^ × π/TL (mm)]}
where A is the horizontal orbital diameter (mm) and B is the vertical orbital diameter (mm). 

Portions of ovaries were fixed in Bouin’s solution for histological observation. The right-side eye was fixed in 4% paraformaldehyde with phosphate-buffered saline (PBS, pH 7.8) for fluorescence in-situ hybridization.

### 2.2. Histological Procedures and Specimen Collection and Classification 

Fixed samples were dehydrated through an ethanol series, embedded in paraffin wax, and sectioned at 7–8 μm thickness. Sectioned tissues were stained with Mayer’s hematoxylin and eosin to evaluate ovarian development (Figure 1 and Appendix A), and development was staged according to the classification in the European eel [24]:

-Stage 1: Ovary is occupied exclusively by oocytes at the peri-nucleolus stage (immature stage; Figure 1a, *n* = 8).

-Stage 2: Yolk is incorporated peripherally into oocytes, in which small yolk globules and oil droplets appear (early vitellogenic stage; Figure 1b, *n* = 13).

-Stage 3: Various sizes of yolk globules distributed throughout the cytoplasm of oocytes, but oil droplets are still seen (mid vitellogenic stage; Figure 1c, *n* = 8).

-Stage 4: The cytoplasm of fully-developed oocytes is occupied by yolk globules (late vitellogenic stage; Figure 1d, *n* = 6).

-Stage 5: The germinal vesicle moves from the center to the peripheral region of oocyte (final maturation stage; Figure 1e, *n* = 6).

Oocyte diameter (OD) at each development stage was determined under a microscope.

### 2.3. Plasma Steroid Hormone Assay

Plasma E_2_ levels were measured by enzyme-linked immunosorbent assay (ELISA) as previously described by Asahina et al. [25]. Each plasma sample (100 μL) was extracted twice using 3 times volume (300 μL) of diethyl ether. An aliquot was transferred to a new microtube and subjected to evaporation (VEC-310, EYELA, Tokyo, Japan). After evaporation, 100 μL of 50 mM borate buffer (pH 7.8, containing 0.5% bovine serum albumin) was added and vortexed. With the exception of blank wells, all wells in a 96-well plate were coated with 100 μL (15 μg/mL) of Affinipure goat anti-rabbit IgG (H+L) (Jackson ImmunoResearch, West Grove, PA, USA) in 50 mM carbonate buffer (pH 9.6) and incubated for 2 h at 4 °C. The wells of the plate were washed three times with 10 mM PBS containing 0.05% Tween (PBS-Tween) using an Immuno Wash 1575 microplate washer (Bio-Rad, Hercules, CA, USA). The assay was performed in a total volume of 150 μL, which contained 50 μL E_2_ standards (Sigma-Aldrich; 12.8–0.025 ng/mL) or plasma samples, 50 μL diluted steroid labeled with horseradish peroxidase (Cosmo-Bio, Tokyo, Japan), and 50 μL rabbit anti-E_2_ antibody (Cosmo-Bio). Incubation was performed for 2 h at 20 °C. After washing three times with PBS-Tween, a 100 μL 0.2 M citrate buffer (pH 4.5) containing 0.01% *o*-phenylenediamine dihydrochloride (Sigma-Aldrich) and 0.04% H_2_O_2_ was added to each well. After incubating the plate in the dark for 30 min at room temperature, 25 μL 4 N H_2_SO_4_ was added to each well to stop the reaction. The absorbance of each well was measured using a microplate reader (PerkinElmer, Waltham, MA, USA) at 450 nm.

### 2.4. RNA Extraction and cDNA Synthesis 

Total RNA was extracted from the whole brain, pituitary, left-side eye, and ovary using RNAiso Plus (Takara Bio, Otsu, Japan) according to manufacturer instructions. To prevent contamination from genomic DNA, total RNA (1 μg) was treated with DNase I (Promega, Madison, WI, USA) at 37 °C for 15 min. The total RNA quantity was measured at 260 and 280 nm, and samples exhibiting an A_260_:A_280_ ratio of 1.8–2.0 were used for cDNA synthesis. cDNA was synthesized using the Transcriptor First Strand cDNA Synthesis Kit according to the manufacturer protocol (Roche Diagnostics, Basel, Switzerland). The cDNA was used as a template for qPCR. 

### 2.5. Real-Time qRT-PCR (qPCR)

Relative ER mRNA expression was analyzed by qPCR, which was performed using a Dice real time thermal cycler (Takara Bio) and SYBR Premix Ex Taq™ II (Takara Bio). PCR primers were designed from the NCBI entries of ERα (GenBank Accession No. HM545084.1), ERβ (AB003356.1), and Ef1α (MH020210) coding regions (Table 1). Each PCR mix contained 0.5 volume of SYBR Premix, 0.2 μM of each forward and reverse primer, and 50 ng of cDNA template. The initial 1-min denaturation was followed by 40 cycles of denaturation for 5 s at 95 °C, annealing, and extension for 1 min at 60 °C. To ensure the specificity of the PCR amplicons, the temperature of the sample was gradually increased from 60 to 95 °C during the last step of PCR, and the melting curve was analyzed. The primers were successfully tested in the various cDNA samples obtained from Japanese eels by evaluating that each primer should amplify a single product, which is reflected as a single peak in the melting curve analysis. Also, the efficiency of the amplification remained between 90 and 110%. The relative mRNA expression levels of target genes were calculated using the ^ΔΔ^Ct method, and the reference gene was virtually defined as the average of the threshold cycles (Ct) for EF1α.

### 2.6. Fluorescence In-Situ Hybridization (FISH) 

For FISH, Stellaris RNA FISH (Biosearch Technologies, Novato, CA, USA) probes were custom-designed for the Japanese eel ERα and ERβ transcripts and labeled with CAL Flour Red 590 according to manufacturer instructions. The fixed right-side eye samples (n = 6) were dehydrated through an ethanol series, embedded in paraffin wax, and sectioned at 7–8 μm thickness. Slides were deparaffinized in xylene and rehydrated in ethanol series. Slides were immersed in 70% ethanol for 1 h at room temperature. After rinsing with 1x PBS for 5 min, slides were immersed in pre-warmed (37 °C) proteinase K solution (10 μg/mL proteinase K in 1x PBS) for 20 min at 37 °C. After rinsing with 1x PBS for 5 min and immersing in wash buffer for 5 min, 100 μL of hybridization buffer containing probe (125 nM) was dispensed onto the tissue section of the slide, and this section was then incubated in the dark at 37 °C for 12 h. After rinsing with wash buffer A for 30 min at 37 °C in the dark, the slide was stained with wash buffer consisting of 5 ng/mL DAPI (Sigma-Aldrich) for 30 min at 37 °C in the dark and then immersed in wash buffer for 5 min. Finally, 50 μL of VectaMount AQ Aqueous Mounting Medium (Vector Laboratories, Burlingame, CA, USA) was added onto the tissue section, and tissues were covered with clean cover glass. 

### 2.7. Statistical Analysis

All statistical analyses were performed using GraphPad prism 8.0.2 Software (version 8.0.2, GraphPad Software, San Diego, CA, USA). One–way analysis of variance (ANOVA) was performed to compare the plasma E_2_ levels, morphological index (GSI, EI), and ERs mRNA expression followed by Tukey multiple comparison test. Data are represented in box-whiskers format, where the band within the box represents the median, and the whiskers represent the minimum and maximum values. A probability of *P* < 0.05 was used in the present study to denote statistical significance.

## 3. Results

### 3.1. Changes in Morphometric Parameters and Plasma E_2_ Levels during Ovarian Development 

Figure 2 shows changes in GSI, OD, and EI in the female Japanese eel after intraperitoneal injections with SPE. When each parameter was arranged according to histological observation of oocyte development within an ovary, GSI was low at Stage 1 (0.18 ± 0.07) and increased significantly (*P* < 0.001) from Stage 3 to Stage 5. Concomitantly, OD and EI significantly (*P* < 0.001) increased toward Stage 5.

Plasma E_2_ levels were low (0.42 ± 0.06 ng/mL) at Stage 1, and a rapid and significant increase in these levels (*p* < 0.01) was observed at Stage 3. High E_2_ levels were then maintained from Stage 3 to Stage 5 (Figure 3).

### 3.2. Changes in ER Transcript Levels within Tissues

After SPE injections, the transcript levels of ERα and ERβ were quantified in several tissues using qPCR. Our results indicated that the transcript profiles of ERα and ERβ were different among tissues, although SPE injections up-regulated their transcription in most tissues. The abundance of ERα mRNA significantly (*P* < 0.01) increased in the eye with a peak at Stage 4, while levels of ERβ mRNA consistently increased toward Stage 5. A steady increase in ERα mRNA transcription was observed in the brain, while levels of ERβ mRNA significantly decreased at Stage 2 but increased to Stage 5. The transcript levels of ERα mRNA in the pituitary increased from Stage 1 to 3 (*P* < 0.01), while levels of ERβ mRNA exhibited little change from Stage 1 to 5. mRNA levels of ERα and ERβ in the ovary remained significantly elevated from Stage 3 to Stage 5 (Figure 4 for ERα and Figure 5 for ERβ and Appendix A). 

### 3.3. Localization and Expression Patterns of ER mRNA in the Japanese Eel Eye

Figure 6 and Figure 7 both illustrate the results of FISH experiments to determine the localization of ERα and ERβ mRNA in the Japanese eel eye. Both ERα and ERβ mRNAs were detected in the choriocapillary layer (CCL) and photoreceptor layer (PRL) of the eye. When the expression patterns were examined using the same method, ERα and ERβ mRNAs were weakly labeled in the CCL of the immature eel eye (Figure 6a and Figure 7a). This result was comparable to the CCL of the mature eel eye, in which ERα and ERβ mRNAs were strongly labeled (Figure 6d and Figure 7d). Conversely, ERα and ERβ mRNA signals in the PRL did not differ between the immature and mature eye (Figure 6a,d and Figure 7a,d).

## 4. Discussion

The results of our present study demonstrated that GSI and OD increased in the SPE-injected females. It is clear that increases in these parameters are related to active vitellogenin synthesis in the liver and its incorporation into developing oocytes within the ovary. This result was in agreement with the previous studies in which significant increases in those parameters were recorded during sexual maturation from yellow to silver eels [19,22]. Similar results have been obtained from experiments performed on European eels [21,26,27]. 

The present histological observation revealed that vitellogenesis (Stage 2 to Stage 4) and oocyte maturation (Stage 5) were induced by hormone injections. As E_2_ increased rapidly from Stage 3, where yolk is actively incorporated into oocytes, this sex steroid was involved in vitellogenesis of the Japanese eel. Similar results were reported in other eels. Specifically, the plasma E_2_ levels in the early- and mid-vitellogenic oocytes of migratory eels were significantly higher than those in pre-vitellogenic oocytes of non-migratory eels [28], and the plasma E_2_ levels were significantly higher in pre-silver eels than in silver eels, where the ovaries of these eels contained oocytes at the peri-nucleolus stage and the oil-drop stage, respectively [29]. This is likely due to the accumulation of vitellogenin within the gonads and subsequent secretion of E_2_ into the body. Similar results were observed in European eels [24,27]. During the spawning migration of longfin eels without hormone treatments, individuals possessing early- to mid-vitellogenic oocytes exhibited significant increases in plasma E_2_ levels [30]. Similar to observations in teleosts, the accumulation of vitellogenin and subsequent increase in E_2_ levels are believed to facilitate sexual maturation in eels. Recently, artificial induction of sexual maturation using androgen-type hormones was attempted in female eels [31,32]. These studies are based on the observation that 11-KT affects initial oocyte development in female eels [33]. The blood content of 11-KT is relatively lower than that of E_2_, however, and given this its involvement is believed to be relatively lower than that of E_2._ Further studies are needed to clarify the role of this hormone.

Species-specific morphometric changes in eels and fish may be determined by endocrine signals accompanied by behavioral changes during catadromous migration; however, the direct and physiological effects of these morphometric changes, particularly the direct effects of increased EI, on the reproductive endocrine system have not been widely reported. A small number of previous studies have suggested that eye size increases to adapt to the deep depths of the spawning grounds, and that these changes are affected by 11-KT [21]; however, the exact mechanisms remain unknown. Among the various morphometric changes that occur during sexual maturation of the *Anguillid* sp., changes in eye size are often used as a quantitative criterion to differentiate between yellow and silver European eels (EI ≤ 6.5) [21]. Additionally, in Japanese eels, an EI ≤ 5.0 is used as a standard to differentiate yellow eels from silver eels [22,23]. While there were some slight differences among the increases in eye size during the sexual maturation of various *Anguillid* sp. in the current study, the close relationship between increased eye size and sexual maturation was confirmed.

Estrogen receptors are cytosolic transducers of estrogen signals within cells or neurons. Therefore, the amount and size of ER mRNA may be related to the function and control of estrogen [10]. In studies of ER in several fish species, the expression trends of E_2_ plasma levels and E_2_ receptors were reported to coincide. ERs in teleosts are generally thought to be controlled by the secretion of E_2_, but trends differ by species, organs, sex, and stages of sexual maturation. The current study evaluated the expression of two ER mRNA subtypes (ERα and ERβ) in response to the changes of plasma E_2_ levels on the BPG (brain-ptuitary-gonad) axis during the sexual maturation of Japanese eels. ERα mRNA expression in female eels with induced sexual maturation via SPE hormone treatment exhibited increasing trends that were directly correlated with E_2_ secretion in all organs; however, ERβ mRNA expression increased only in stage 3 within the ovaries and remained high until Stage 5. This is likely due to the effects of up-regulation of ERα on the BPG axis and ERβ in the ovaries during sexual maturation in female eels. This suggests that E_2_ secretion during the sexual maturation of female Japanese eels is directly related to ERα on the BPG axis, and that ERα and ERβ play different roles in the sexual maturation process. Similar to our current results, Jeng et al. [34] reported that plasma E_2_ levels are responsible for the up-regulation of ESR-α mRNA expression, and not that of ESR-β, in the brain. Additionally, during the process of inducing sexual maturation in female European eels through carp pituitary homogenate treatment, ESR-α mRNA expression levels in the brain and pituitary gland increased significantly, while those of ESR-β did not significantly change. Generally, in studies of ERs in teleosts, the expression trends of ERs in response to the secretion of E_2_ differ between developmental stages of the gonads and different organs. For example, the expression of ESR1 mRNA in the ovaries of various teleosts, including goldfish [35], orange-spotted grouper [8], and rainbow trout [13], increased significantly with gonad development. The ERα expression in the ovaries of Asian swamp eels [36] and orange-spotted grouper [8] increased with gonad development. Additionally, the expression of ESR1 mRNA in the brain, pituitary gland, liver, and ovaries of female European eels with artificially induced sexual maturation were significantly greater than expression levels observed in the control group [11]; however, the expression of ER alpha and ER beta I in the livers of female goldfish in response to E_2_ treatments increased [12]. ERs were also found in the brains of medaka, and the expression of these receptors was reported to be affected by reproductive and diurnal cycles [37]. Based on these results, in Japanese eels, the secretion of E_2_ in the bloodstream is closely related to ERα in the BPG axis. ESR1 plays a fundamental role in mammalian reproduction, but information on the distinct roles of each ESR subtype in non-mammalian vertebrates is limited. For example, estrogen induces hepatic vitellogenin production in animals with ovaries, but it is unclear whether ESR subtypes are involved in this process. Recently, CRISPR/Cas9-mediated knock out (KO) of ESR1 was reported to have no effect on the gonadal development and function in female and male esr1 KO zebrafish [38]. Thus, ESR1 may not be directly involved in gonadal development. Anguillid eels have not been suitable for laboratory research because artificial maturation or seed production technology has not been developed yet. However, since European [39] and Japanese eels [40], which were used in this study, are Anguillidae, it will be possible to explain the specific molecular biological mechanisms of ERs after the advent of research using gene regulation technology. As an important factor controlling the neuroendocrine system, E_2_ controls the BPG axis in the brain (for review see [41]). The sexual maturation process of female Japanese eels has been previously reported and, based on this information, our current study investigated not only the expression characteristics of ERs in the BPG axis, but also the expression patterns of ER mRNA within the eel eye. Our results indicated an increasing pattern of blood E_2_ levels during the induction of sexual maturation that paralleled that of ERα mRNA in the eye, but the expression of ERβ mRNA gradually increased with sexual maturation and only reached a significantly high level at stage 5, which occurs immediately prior to spawning. Thus, the eyes of Japanese eels may play an important role in their sexual maturation. In fish, the presence of ER in the eye was reported in goldfish and rainbow trout, and expression of ESR, ESR2a, and ESR2b mRNA in the eyes of zebrafish was confirmed [42,43,44]. In particular, the expression of ESR1 mRNA in the eye was reported to increase upon E_2_ exposure, but the mechanisms underlying sexual maturation and other physiological aspects were not examined. Additionally, ESR1, ESR2, and G-protein coupled estrogen receptor expression in the eyes of European eels during their silvering stage was recently confirmed, but the physiological mechanisms remain unclear. Studies of the roles of ERs in fish eyes and their chemical control mechanisms are limited. In regard to the retinal changes in eels, morphometric changes in the retinal structure and cone density during the silvering processes of European eels were reported, but endocrine studies have not been conducted. The EI was high in female Japanese eels treated with 11-KT l [31], and thus 11-KT was suggested to affect the increase in eye size and initial oocyte development. The gonads, however, were in the oil droplet stage (Stage 2 in the current study) when the experiment ended; this is prior to the yolk accumulation stage, making it difficult to determine the relationship between eye size and sexual maturation. Previous studies conducted in humans support the results of the current research. Studies of the human eye have determined the detailed physiological functions of ERs. According to Kobayashi et al. [45], ER mRNA expression was observed by fluorescence in-situ hybridization on the choroidal neovascular membrane. Additionally, highly myopic eyes were reported to be more prevalent in females than in males, suggesting that estrogen expedites cell proliferation during the formation of choroidal neovascular membranes in highly myopic human eyes. ERα mRNA was also detected in the human eye, and mRNA expression differed with gender and age, indicating the involvement of estrogen [46]. Thus, ERα in the eye is related to the secretion of E_2_ during the induction of sexual maturation in eels, and the eye may be an important organ in reproductive endocrinology. Additionally, ERα and ERβ in the eye are thought to exert different actions in the sexual maturation process; however, further studies are needed to draw the same conclusions in teleost.

We also examined if increased EI directly or indirectly affects E_2_. Using the FISH method, the location and degree of ER mRNA expression in the eye of Japanese eels were detected, and our results indicated that both ERα and ERβ mRNA were present in the PRL and CCL, and the mRNA expressed in the CCL differed between sexually immature and mature individuals. In particular, there was a clear difference between the size and amount of capillary endothelial cells in the CCLs of sexually immature and mature individuals. This suggests that E_2_ is involved in the proliferation of capillary endothelial cells in the eye. Studies of the expression distribution of ER mRNA in mammalian retina reported the presence of RPE in the retina and vessels in the choroid [46,47,48]. Interestingly, E_2_ in the human eye reportedly exerted an important effect on choroidal neovascularization and controlled capillary endothelial cell growth in the retina through a receptor-mediated pathway [45]. These results support those of the current study; however, studies examining the physiological mechanisms of ERs on the sexual maturation of organisms, including fish, are limited. In regard to the differences in sexual maturation processes in *Anguillid* sp., additional studies are needed to determine the specific correlations of the eye and reproductive mechanisms.

In conclusion, blood E_2_ levels increased with ovary development, and this caused the ERα mRNA content to increase accordingly in the BPG axis. Morphometric changes, such as those in EI, were also observed. Additionally, we found that blood E_2_ levels affect the entire development process of female ovaries, supporting the results of previous studies examining female eels [24,27,28,29,30]. Detailed studies investigating the entire ovary development process, however, remain limited. This is also the first study to demonstrate that ER mRNA expression increases in the eye in conjunction with increases in eye size during the sexual maturation of female eels, and, given this, E_2_ may affect the proliferation of capillary endothelial cells in the CCL. Based on these results, eel eyes play a significant role in sexual maturation. Further studies are needed to determine the correlation between the BPG axis and the eye at each developmental stage, and further investigation of the specific cytophysiological signals activated following estrogen-mediated stimulation of the retina is also required.

## Figures and Tables

**Figure 1 cells-08-00310-f001:**
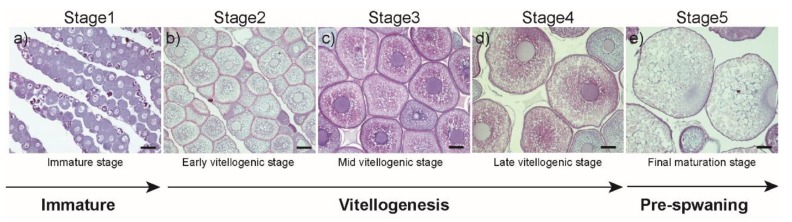
Histological section of oocytes at different development stages during salmon pituitary extract (SPE) treatment. (**a**) Immature stage, (**b**) Early vitellogenic stage, (**c**) Mid vitellogenic stage, (**d**) Late vitellogenic stage, (**e**) Final maturation. Scale bar = 100μm.

**Figure 2 cells-08-00310-f002:**
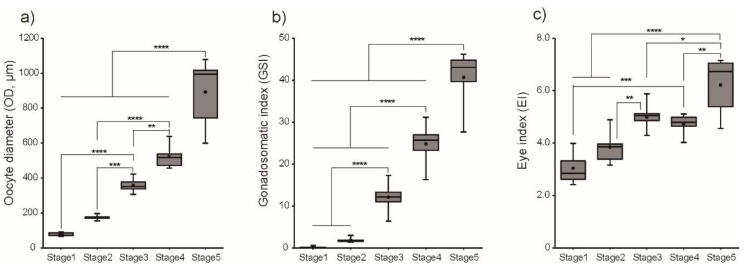
Changes in morphometric parameters, (**a**) Gonadosomatic index (GSI), (**b**) oocyte diameter (OD), and (**c**) eye index (EI), in female eel during artificial maturation. Boxplots show min and max values (whiskers), first and third quartiles (box limits), and median (box inner line). Dots represent mean values of GSI, OD, and EI. Asterisks mean significantly different (*, *p* < 0.05; **, *p* < 0.01; ***, *p* < 0.001; ****, *p* < 0.0001).

**Figure 3 cells-08-00310-f003:**
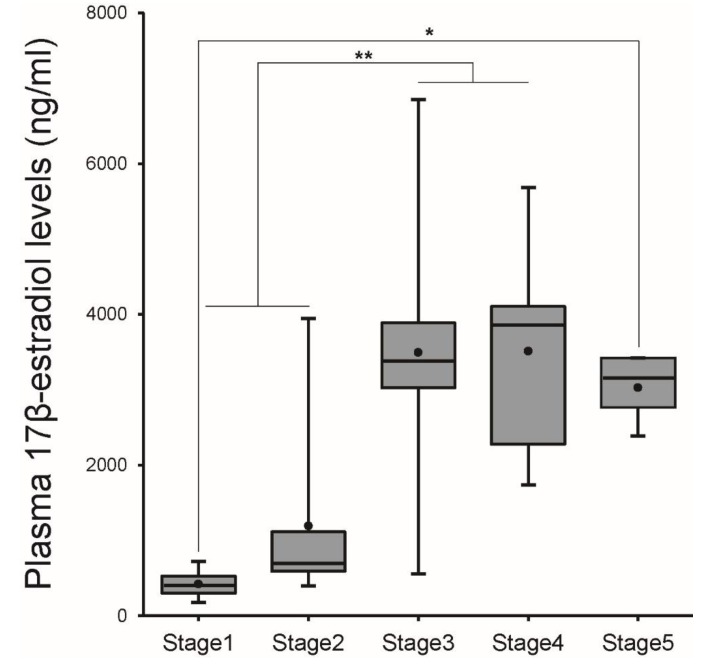
Plasma levels of E_2_ in female eel during artificial maturation. Boxplots show min and max values (whiskers), first and third quartiles (box limits), and median (box inner line) of plasma levels of E_2_. Dots represent mean values of plasma levels of E_2_. Asterisks mean significantly different (*, *p* < 0.05; **, *p* < 0.01).

**Figure 4 cells-08-00310-f004:**
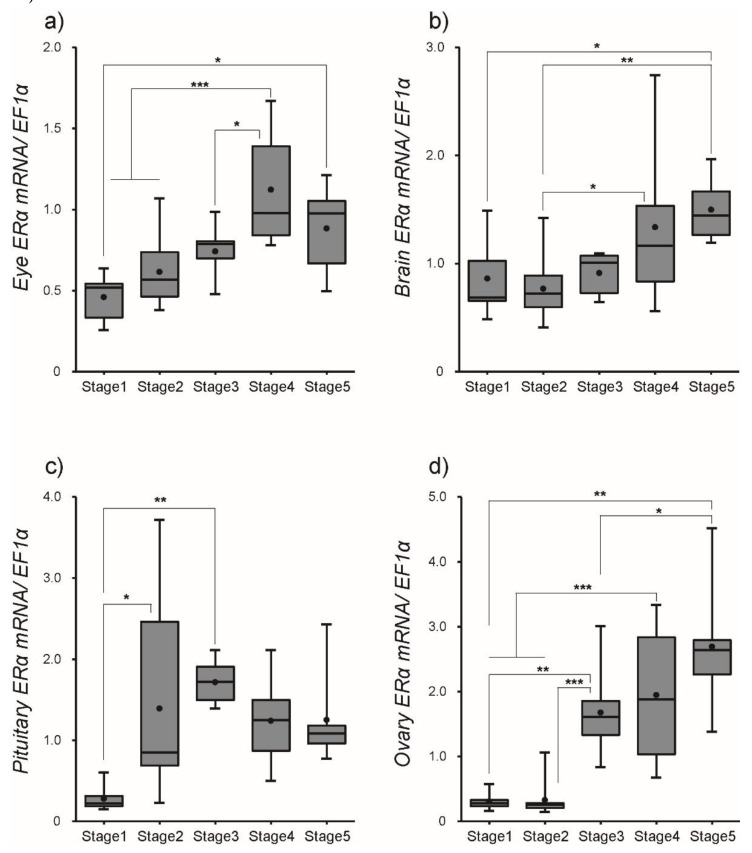
The mRNA expression of estrogen receptor (ER)α in female eel in (**a**) eye, (**b**) brain, (**c**) pituitary, and (**d**) ovary during artificial maturation as measured by real-time quantitative PCR. Boxplots show min and max values (whiskers), first and third quartiles (box limits), and median (box inner line) of mRNA expression levels. Dots represent mean values of mRNA expression levels. Asterisks mean significantly different (*, *p* < 0.05; **, *p* < 0.01; ***, *p* < 0.001).

**Figure 5 cells-08-00310-f005:**
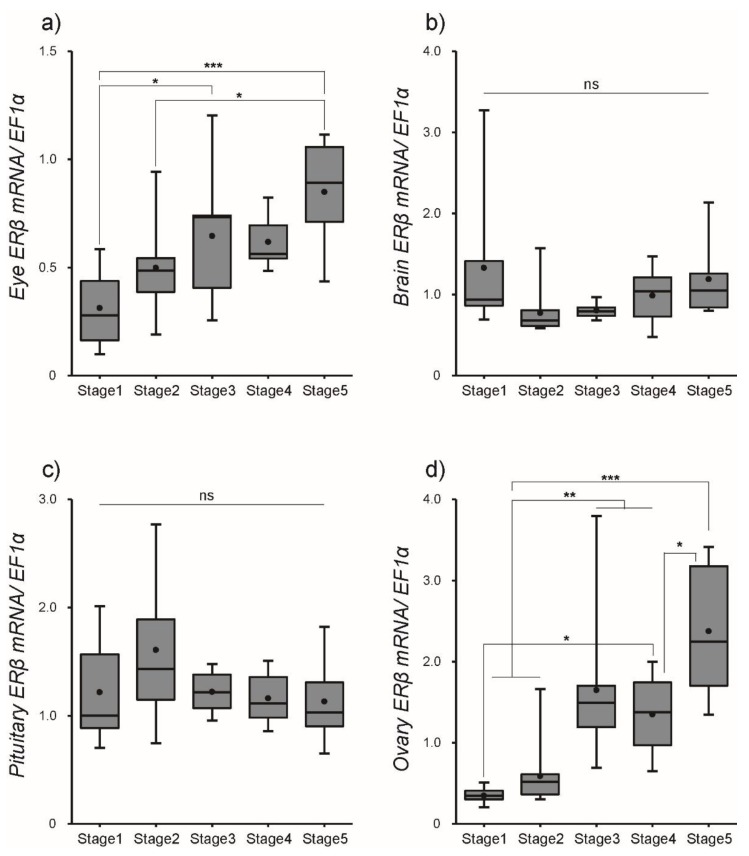
The mRNA expression of ERβ in female eel in (**a**) eye, (**b**) brain, (**c**) pituitary, and (**d**) ovary during artificial maturation as measured by real-time quantitative PCR. Boxplots show min and max values (whiskers), first and third quartiles (box limits), and median (box inner line) of mRNA expression levels. Dots represent mean values of mRNA expression levels. Asterisks mean significantly different (*, *p* < 0.05; **, *p* < 0.01; ***, *p* < 0.001; ns, no significant difference).

**Figure 6 cells-08-00310-f006:**
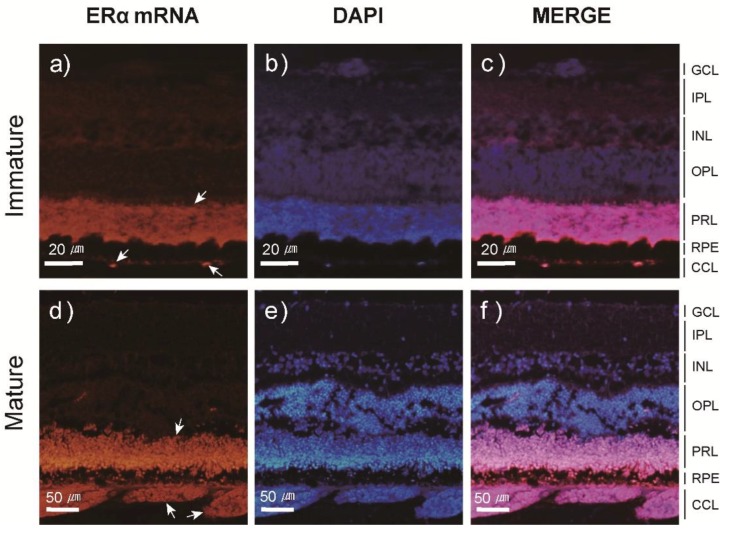
Fluorescence in situ hybridization of ERα mRNA in sexually immature (**a**) and mature (**d**) eye sections. Nuclei were stained with DAPI (**b**, **e**). A merge of the two different channels (**c**, **f**). The arrow indicate the expression of ERα mRNA in Japanese eel eye. Abbreviations: Choriocapillary layer (CCL), ganglion cell layer (GCL), inner nuclear layer (INL), inner plexiform layer (IPL), outer plexiform layer (OPL), photoreceptor layer (PRL), retina pigment epithelium (RPE).

**Figure 7 cells-08-00310-f007:**
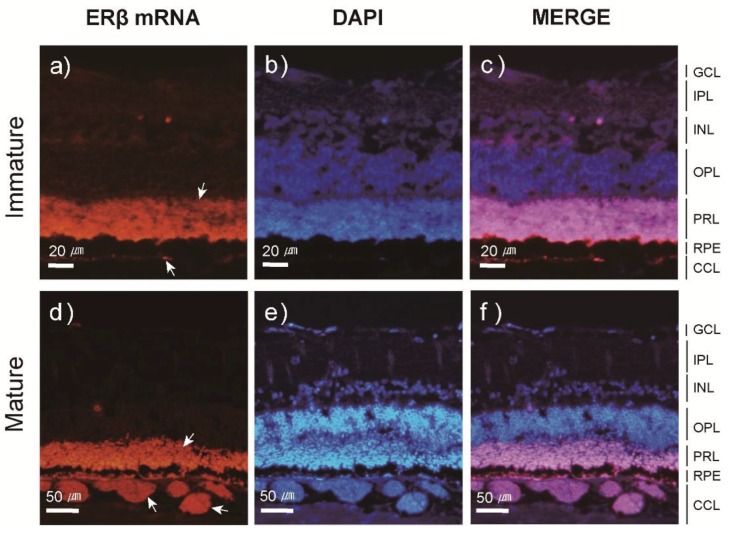
Fluorescence in-situ hybridization of ERβ mRNA in sexually immature (**a**) and mature (**d**) eye sections. Nuclei were stained with DAPI (**b**, **e**). A merge of the two different channels (**c**, **f**). The arrow indicate the expression of ERβ mRNA in Japanese eel eye. Abbreviations: Choriocapillary layer (CCL), ganglion cell layer (GCL), inner nuclear layer (INL), inner plexiform layer (IPL), outer plexiform layer (OPL), photoreceptor layer (PRL), retina pigment epithelium (RPE).

**Table 1 cells-08-00310-t001:** Primers used in quantitative polymerase-chain reaction (PCR) analysis.

Gene (Accession No.)	Primer	Sequence	Product Size (bp)
Ef1α (MH020210)	Forward	5′-TCACCCTGGGAGTAAAGCAG-3′	222
	Reverse	5′-TCCATCCCTTGAACCAGGAC-3′	
ERα (HM545084)	Forward	5′-TGATCGCTTGGGCTAAGAAAGT-3′	209
	Reverse	5′-GTCGAAAATCTCGGCCATGC-3′	
ERβ (AB003356)	Forward	5′-AAGTACACCTGCTGGAGTGC-3′	220
	Reverse	5′-AGGCACACATACTCCTCCCT-3′	

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
