# Peer review of "Involvement of Estrogen and Its Receptors in Morphological Changes in the Eyes of the Japanese Eel, Anguilla japonica, in the Process of Artificially-Induced Maturation"

_cells, 2019, doi:10.3390/cells8040310_

Round 1
Reviewer 1 Report
Comments for the Author:
The authors presented the blood E2 levels increased with ovary maturation and eye morphometric changes. PIs performed the artificial induced maturation to induce Japanese eel ovary maturation. The authors showed the blood E2 levels increased with ovary development however, detailed molecular pathway is limited. ERα mRNA content was increased accordingly with gonadosomatic index and eye index. Increases in the eye size during the sexual maturation of female eels, and concluded that eel eyes play a significant role in sexual maturation.
1. No supported data was shown to identify the eel eyes play a significant role in sexual maturation. Most results were found in association. Authors should performed the eye-specific knock out eel or ESR1 knock out or ESR2 knock out eel to clarify the role of ESR1 in Brain-Pituitary-Gonad pathway and eye maturation, ESR2 in the ovary development.
2. Recently, several literatures discussed about the generation of gene knockout fishes or eel with TALENs should be presented and discussed.
3. All the mRNA data should provide the statistic information and detail information about the numbers.
4. ESR-α (Line 306) and ESR-β (Line 307) should be corrected as ESR1 and ESR2.
Author Response
Response to Reviewer 1 Comments
Point 1. No supported data was shown to identify the eel eyes play a significant role in sexual maturation. Most results were found in association. Authors should performed the eye-specific knock out eel or ESR1 knock out or ESR2 knock out eel to clarify the role of ESR1 in Brain-Pituitary-Gonad pathway and eye maturation, ESR2 in the ovary development.
Response 1: This study aimed to elucidate the relationship between estradiol (E2) and the unexplained phenomenon in which the eyes of Anguillid eels enlarge during sexual maturation. To the best of our knowledge, no previous study has investigated the phenomenon in Anguillid eels or its relationship with the reproductive axis. Salmon pituitary extract (SPE) is the only known way to induce sexual maturation in Anguillid eels, including the Japanese eels used in this study, and the underlying mechanisms or characteristics of the 2,3-bisphosphoglyceric acid pathway have not been reported in Anguillid eels. Moreover, although SPE treatment may result in fertilized eggs, successfully hatching of these eggs is rare. Further, since an appropriate diet is not yet available for the larvae, it is currently impossible to apply knockout technology to Anguillid eels. If seed production methods are developed in the future, we agree that the study that you suggested should be conducted.
Point 2. Recently, several literatures discussed about the generation of gene knockout fishes or eel with TALENs should be presented and discussed.
Response 2: According to this comment, we added more details explain in discussion (Line 318-328).
Point 3. All the mRNA data should provide the statistic information and detail information about the numbers.
Response 3: According to this comment, we added detail statistic information in figure caption and supplement date, respectively.
Point 4. ESR-α (Line 306) and ESR-β (Line 307) should be corrected as ESR1 and ESR2.
Response 4: We have changed ESR1 and ESR2 to ESR-α and ESR-β according to the comments by the reviewer.
Reviewer 2 Report
This manuscript titled, "Involvement of estrogen and its receptors in 2 morphological changes in the eyes of the Japanese 3 eel, Anguilla japonica, in the process of artificially-4 induced maturation" investigates the role of the estrogen receptors (ERs) and E2 in the sexual maturation of the Japanese eel (Anguilla japonica) with a specific reference to ERs in the eye. Overall, the manuscript demonstrates clarity in rationale, experiment design, and interpretation of data. As such, I recommend the publication of this manuscript, pending a few changes:
Major changes:
I think the manuscript is relevant, given that estrogen receptor characterization in teleosts like the zebrafish have paved the way for the investigation of basic and translational research problems alike. However, I think the mere identification of receptor transcripts in the eel eye may not constitute a well-rounded manuscript, without any further investigation of downstream estrogen targets. Therefore, it is recommended that the authors execute gene expression analyses of estrogen receptor target genes directly related to their study (such as vtg) to prove the activation of the estrogen receptors. This will further strengthen the conclusions of the manuscript.
Minor changes:
In Figures 2, 3, 4A and $B, it is not clear as to which experimental groups were compared for statistical analyses. Therefore, it is recommended that the authors use segmented lines or other symbols to indicate the groups being compared. Please also mention the p values directly instead of a, b, c etc, or use conventional asterisks (*) symbology, as it is makes it these tests easier to interpret.
Line 30, please remove the word "Since".
Line 61, a preposition is missing in "Mariana Islands the Mariana Trench"
Author Response
Response to Reviewer 2 Comments
Point 1: I think the manuscript is relevant, given that estrogen receptor characterization in teleosts like the zebrafish have paved the way for the investigation of basic and translational research problems alike. However, I think the mere identification of receptor transcripts in the eel eye may not constitute a well-rounded manuscript, without any further investigation of downstream estrogen targets. Therefore, it is recommended that the authors execute gene expression analyses of estrogen receptor target genes directly related to their study (such as vtg) to prove the activation of the estrogen receptors. This will further strengthen the conclusions of the manuscript.
Response 1: To verify the activation of estrogen receptors, as suggested by the reviewer, we aim to analyze the expression of target genes of estrogen receptors that are directly related to vitellogenesis. Although estrogen certainly induces vitellogenin production in the ovaries, the goal of the present study is to investigate the influence of E2 produced during sexual maturation on the eyes. E2 is secreted during sexual maturation. This study particularly aimed to discuss the role of E2 in the enlargements of the eyes and retina, independent of the canonical estrogen receptor pathway. Significant enlargement of the eyes during sexual maturation in vertebrates, including fish, is a very special phenomenon. Therefore, we hypothesized that this morphological change is based on E2 and proceeded with our experiments. Seed production was impossible in the Japanese eels used in this study, and more than 95% of juvenile fish captured and grown under farming conditions are male, for some unknown reason. Accordingly, females that are used need be captured from the natural environment. Since the Japanese eel is classified as a threatened species by the International Union for Conservation of Nature, only eels captured between September and October may be used. This would extend the time required for future studies. To address the reviewer's comments, we plan to perform these experiments as part of a future study.
Minor changes:
Point 2: Figures 2, 3, 4A and $B, it is not clear as to which experimental groups were compared for statistical analyses. Therefore, it is recommended that the authors use segmented lines or other symbols to indicate the groups being compared. Please also mention the p values directly instead of a, b, c etc, or use conventional asterisks (*) symbology, as it is makes it these tests easier to interpret.
Response 2: We have also added * symbols and P-values directly to the figures. However, to improve the clarity of the figures, we additionally used alphabetical letters for multiple comparisons. As suggested by the reviewer, we also added an additional explanation in the figure caption.
Point 3: Line 30, please remove the word "Since".
Response 3: According to reviewer comment, we delete a “since” in the sentence.
Point 4: Line 61, a preposition is missing in "Mariana Islands the Mariana Trench"
Response 4: According to reviewer comment, we changed “the Mariana Islands the Mariana Trench” to “the Marian Ridge”.
Round 2
Reviewer 1 Report
All the comments have been response adequately.
Reviewer 2 Report
Since the authors were able to address/respond satisfactorily to my queries/concerns, I now recommend this article for publication.